# An Integrated Vibration Elimination System with Mechanical-Electrical-Magnetic Coupling Effects for In-Wheel-Motor-Driven Electric Vehicles

Ze Zhao [1,*], Liang Gu [1], Jianyang Wu [2], Xinyang Zhang [3,4] and Haixu Yang [3,4]

[1]  School of Mechanical Engineering, Beijing Institute of Technology, Beijing 100081, China
[2]  Beijing Institute of Space Launch Technology, Beijing 100076, China
[3]  School of Mechanical Engineering, University of Science and Technology Beijing, Beijing 100083, China
[4]  Shunde Innovation School, University of Science and Technology Beijing, Foshan 528399, China
*   Correspondence: zhaoze1990@bit.edu.cn

**Abstract:** This study aims to improve the vehicle vertical dynamics performance in the sprung and unsprung state for in-wheel-motor-driven electric vehicles (IWMD EVs) while considering the unbalanced electric magnetic force effects. An integrated vibration elimination system (IVES) is developed, containing a dynamic vibration-absorbing structure between the IWM and the suspension. It also includes an active suspension system based on a delay-dependent $H\infty$ controller. Further, a novel frequency-compatible tire (FCT) model is constructed to improve IVES accuracy. The mechanical-electrical-magnetic coupling effects of IWMD EVs are theoretically analyzed. A virtual prototype for the IVES is created by combining the CATIA, ADAMS, and MatLab/Simulink, resulting in a high-fidelity multi-body model, validating the IVES accuracy and practicability. Simulations for the IVES considered three different suspension structure types and time delay considerations were performed. Analyses in frequency and time domains for the simulation results have shown that the root mean square of sprung mass acceleration and the eccentricity are significantly reduced via the IVES, indicating an improvement in ride comfort and IWM vibration suppression.

**Keywords:** in-wheel-motor; unbalanced electric magnetic force; vertical-longitudinal dynamics; road-tire-rotor force; multi-optimization method; virtual prototype

## 1. Introduction

Automotive electrification is rapidly expanding worldwide to tackle the challenges such as greenhouse gas emissions and fossil oil depletion. In-wheel-motor-drive electric vehicles (IWMD EVs), in which the drive motor is integrated directly into the wheel, have several benefits, including compactness, controllability, and efficiency. Recently, IWMD EVs started to attract researchers, and are an important research direction for future electric vehicles [1,2]. However, the use of wheel motors increases the unsprung mass, negatively affecting the vehicle dynamic performance, including ride comfort and road holding performance [3,4]. On the other hand, increasing expectations for improved noise-, vibration-, and harshness- (NVH) performance make it more important to deeply investigate correlated characteristics [5,6]. Additionally, IWM vibration might cause motor bearing wear and magnet gap eccentricity [7]. Further, eccentricity can cause an unbalanced electric magnetic force (UEMF) which further distorts the air gap distribution, exacerbating the in-wheel-motor (IWM) vibration, creating a vicious cycle [8]. This is known as a typical mechanical-electrical-magnetic coupling system where UEMF has a key role [9,10].

Numerous studies were carried out to mitigate the adverse coupling effect on vehicle dynamics, and can generally be grouped into two categories. The first category includes designing the IWM as a dynamic vibration absorbing structure (DVAS), where the IWM is isolated from the axle by spring and damper elements [11]. The DVAS absorbs the vibration

energy transmitted to the IWM by damper element [12]. Further, DVAS is generally divided into "chassis DVAS" and "tire DVAS", which flexibly connects the IWM to the sprung and unsprung masses, respectively [11]. Previous studies have shown that DVAS can partially suppress the IWM vibrations if the parameters of the additional spring dampers are properly selected [13,14]. Furthermore, an active DVAS control method is proposed in [15] through the installation of a controllable linear motor between the IWM stator and the wheel axle, effectively reducing the wheel vibration. Most existing DVAS installed in the IWM are passive vibration absorbers with fixed parameters, which cannot be adjusted to complex and variable road excitations. On the other hand, the active DVAS can achieve better performance but is seldom applied due to its cost and space constraints. In addition, when improving IWM vibrations, DVAS has difficulty optimizing sprung mass vibrations in the 4–8 Hz frequency range of the passenger-sensitive [16].

The second category considers the IWM suspension as a complete system, which utilizes conventional active suspension system (ASS) control algorithms to reduce the negative impacts of the increase in unsprung mass. Examples of such algorithms are explicit model predictive control method [17], fuzzy logic control [18], ceiling damping control [19], optimal sliding mode [20], and $H\infty$ control [21]. Active suspension systems inevitably have time delay, which is one of the most important factors affecting the stability of the system. Among the above control strategies, $H\infty$ control is widely used since it provides increased system robustness for time delay, and time constant perturbation [22]. Sun et al. [23] designed an adaptive robust $H\infty$ controller for electro-hydraulic actuator active suspension, robust to the uncertainty of actuator parameters and its nonlinearity. In [24], the robust $H\infty$ controllers were designed for ER suspension systems with parameter uncertainties. The dynamic bandwidth of the current-variable damper under the fast and slow response action was determined. Li et al. [25] proposed using a robust $H\infty$ control method to improve vehicle performance under load variations. The required vehicle suspension properties such as ride comfort, car handling, and suspension deflection are transformed into a continuous time system with input delay and sector bounded uncertainty. Shao et al. [26] studied the $H\infty$ control design method for an ASS of IWMD EVs while considering the actuator failure, time delay, and disturbances. Jing et al. [27] designed an ASS controller for IWMD EVs, aiming to isolate forces transmitted to the motor bearings and improve ride comfort. However, the ASS is a complex system with multiple parameters, and the mechanical-electrical-magnetic coupling effects will further aggravate this condition. Moreover, due to the time delay and power limitation of the actuator, the adjustable frequency range of the ASS is concentrated in the low and mid frequencies [28].

The DVAS and ASS significantly reduce the vibration of unsprung and sprung vehicle masses, respectively. However, few studies combined the ASS control strategies and DVAS approaches to mitigate the mechanical-electrical-magnetic coupling effects. Liu et al. [29] proposed a two-stage optimal control method to improve vehicle dynamics performance through ASS and DVAS. A linear quadratic regulator controller based on the particle swarm optimization and finite frequency $H\infty$ controller are designed for DVAS and ASS, respectively. However, the UEMF of IWM and the time delay of the ASS were not considered. Wang et al. [30] explored the $H\infty$ control strategy for an ASS system in the IWMD EVs, which reduced the IWM vibration and improved the ride comfort. However, the IWM UEMF was not considered. Liu et al. [31] used the particle swarm optimization algorithm to optimize the DVAS parameters, while the alterable-domain-based fuzzy control method was used to control the DVAS actuator force. However, the proposed integration structure was limited by its complexity and was not effectively validated. Since DVAS and ASS have different roles in the IWMD EVs, their combination strategies should be studied to improve motor and suspension performances.

Generally, simplified tire models including only the spring characteristics are used in vehicle vertical control; however, the increase in unsprung mass and the high-frequency excitation generated by the motor directly affect tire working conditions [32]. In particular, the increase in the unsprung mass causes an increasingly non-linear relationship between



the contact force and the road roughness [33]. Typical tire models include physical and empirical tire models. The brush and ring models are two main forms of physical models; the magic formula tire model is attributed as an empirical tire model. However, brush and magic formula tire models are used for modeling the tangential tire force characteristics, not applicable for calculating vertical tire force [34,35]. The most commonly used models in vertical tire dynamics calculation is ring model, which is typically represented by the rigid ring model (RRM) [36] and the flexible ring model (FRM) [37]. For the former, the residual stiffness is introduced between the contact patch and the rigid ring to represent the static tire stiffness in the vertical directions. However, the tire belt deformation was not considered and the applicable frequency range during the analysis is usually low [36]. Further, the FRM uses a large number of segments interconnected by springs and dampers. The FRM bandwidth is thus up to 150 Hz, corresponding to the first flexible belt bending modes; however, it could rarely respond to low-frequency characteristics [37]. As to the IWMD EVs, there are complex maneuvers including high-frequency (50–100 Hz [38]) vibration of the IWM system, as well as low frequency excitations (under 20 Hz) from the road unevenness. Both groups exert serious influences on the vertical dynamics of the vehicle. Consequently, to improve the accuracy and adaptability, it is necessary to integrate the RRM and FRM.

As described above, the combination of DVAS and ASS should be comprehensively investigated to improve the vehicle vertical performance; the UEMF created by mechanical-electrical-magnetic coupling effects should also be considered. Additionally, an accurate road-tire-IWM model is necessary to simulate the real external excitation of the IWM suspension system. Finally, the scientific contributions of this study are:

1. A novel IVES was developed, containing a practical DVAS equipped between the IWM and the suspension, as well as a robust ASS based on the delay-dependent $H\infty$ controller. The UEMF was considered in this system and the mechanical-electrical-magnetic coupling effects of IWMD EVs were observed.
2. A frequency-compatible tire (FCT) model integrating the RRM and FRM was developed to ensure adaptability to different frequency ranges. It also improved the accuracy of vertical tire forces, which were further inputted as an external excitation to the IVES (instead of the road roughness), further improving the accuracy.
3. A novel virtual prototype was designed, combining CATIA, ADAMS, and MATLAB/Simulink environment to establish a high-fidelity multi-body model for the IVES. Particularly, the IVES structure was developed, aiming to maximize its integration ability and minimize the impact on the original chassis structure.

The remainder of the paper is organized as follows: the IVES mathematical model combined with the UEMF model, vertical vibration model, and driving model were introduced in Section 2. The proposed delay-dependent $H\infty$ controller was outlined in Section 3. In Section 4, performed simulations and validations were described for the IVES in a virtual prototype. Finally, study conclusions are given in Section 5.

## 2. The Mathematical Model of SIWMS

The IVES mathematical model was developed, containing the UEMF model in driving conditions, an FCT model, and a DVAS-ASS integrated model. The investigated UEMF and FCT model tire force are imported into the DVAS-ASS integrated model as internal and external excitation, respectively.

### 2.1. The UEMF Model in Driving Conditions

For a switched reluctance motor (SRM), the electromagnetic force can be decomposed into tangential and radial forces according to the electrical angle [39]. The material of stator and rotor is ferromagnetic, the radial force of electromagnetic force is much larger than the tangential force [40,41], and the eccentricity of the rotor and stator of IWM will cause a large deviation of radial force, which acts on the stator of the IWM to produce a violent vibration. From the above analysis, it is evident that the UEM is produced by the coupled action of electromagnetic and mechanical fields and represents the resultant global magnetic force

acting on the rotor and stator due to an asymmetric magnetic field distribution in the air gap, The UEMF generation process is shown in Figure 1. In the figure, $F_t$ is the tangential force, $F_r$ is the radial force, $F_e$ is the electromagnetic force.

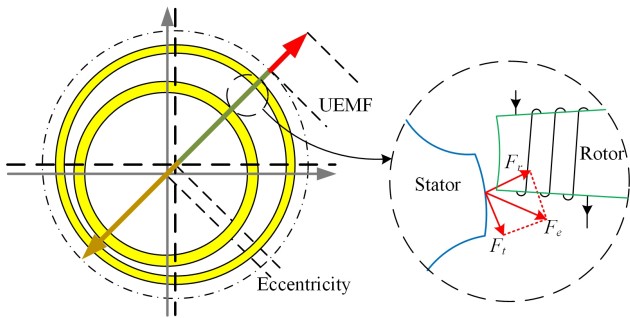

**Figure 1.** UEMF generated in a SRM.

In this paper, a 5-kW exterior rotor switched reluctance motor (SRM) prototype with 8/6-four phases was used [42]. The magnetic co-energy $W(i, \theta)$ is determined according to the current $i$ and the phase inductance $L(\theta, i)$, where $\theta$ is the rotor angle. The first three terms of $L(\theta, i)$ Fourier expansion are given by:

$$L(\theta, i) = L_0(i) + L_1(i)\cos(N_r\theta + \pi) + L_2(i)\cos(2N_r\theta + 2\pi) \tag{1}$$

where $L_0$, $L_1$, and $L_2$ are calculated as:

$$\begin{cases} L_0(i) = \frac{1}{2}\left[\frac{1}{2}(L_a(i) + L_u) + L_m(i)\right] \\ L_1(i) = \frac{1}{2}(L_a(i) - L_u) \\ L_2(i) = \frac{1}{2}\left[\frac{1}{2}(L_a(i) + L_u) - L_m(i)\right] \end{cases} \tag{2}$$

where $L_a$, $L_u$, and $L_m$ are inductances at fully-aligned ($\theta = 30°$), unaligned ($\theta = 0°$), and intermediate positions, respectively. These parameters can be fitted with polynomials based on either the finite element analysis or the experiment. Considering the relationship between the flux and the inductance, the $k$-th phase flux linkage can be derived as:

$$\begin{aligned} \psi(\theta, i_k) &= \int_0^{i_k} L(i_k, \theta)\mathrm{d}i_k \\ &= \frac{1}{2}\left[\cos^2(N_r\theta) - \cos(N_r\theta)\right]\sum_{n=0}^{N} c_n i^n + \sin^2(N_r\theta)\sum_{n=0}^{N} \mathrm{d}_n i^n \\ &\quad + \frac{1}{2}L_u i_k\left[\cos^2(N_r\theta) + \cos(N_r\theta)\right] \end{aligned} \tag{3}$$

where $c_n = a_{n-1}/n$ and $d_n = b_{n-1}/n$. According to Faraday's law, the phase voltage is:

$$U_k = R_k i_k + \frac{\mathrm{d}\psi_k}{\mathrm{d}t} = R_k i_k + L_k(\theta, i_k)\frac{\mathrm{d}i_k}{\mathrm{d}t} + \omega_t\frac{\partial\psi_k}{\partial\theta} \tag{4}$$

where $_t$ is the angular rotor velocity. The phase current can be written as:

$$i_k = \int \frac{U_k - R_k i_k - \omega_t\frac{\partial\psi_k}{\partial\theta}}{L_k(\theta, i_k)}\mathrm{d}t. \tag{5}$$

For constant phase current $i$, relationships between the magnetic co-energy $W(i, \theta)$, torque $T$, and radial force $F_r$ are defined as:

$$T = \left.\frac{\partial W(\theta, i)}{\partial\theta}\right|_{i=\text{const}}, F_r = \left.\frac{\partial W(\theta, i)}{\partial l_g}\right|_{i=\text{const}} \tag{6}$$

where $l_g$ is the air gap between the rotor and the stator. The phase torque is found using:

$$T_k = \left.\frac{\partial W(\theta,i)}{\partial \theta}\right|_{i=\text{const}} = \int_0^{i_k} \frac{\partial \psi(\theta,i_k)}{\partial \theta} \, di_k$$
$$= \sin(N_r\theta) \sum_{n=1}^{N} \frac{1}{n} e_{n-1} i_k^n + \sin(2N_r\theta) \sum_{n=1}^{N} \frac{1}{n} f_{n-1} i_k^{n_k} \tag{7}$$

where both $e$ and $f$ are intermediate functions. The former, $e$, can be calculated as $e_n = (1/2) N_r c_n$, $e_0 = 0$, and $e_1 = (1/2) N_r (c_1 - L_u)$. The latter, $f$, is given by $f_n = N_r d_n - e_n$, $f_0 = 0$, and $f_1 = (1/2) N_r (2d_1 - c_1 - L_u)$.

Based on the phase torque $T_k$, the radial force of the $k$-th phase can be calculated as [11]:

$$F_{rk} = -\frac{T_k \delta}{l_g}, \tag{8}$$

where $\delta$ is the overlap between the rotor and the stator. The presence of non-zero $l_g$ would result in the UEMF. There are many possible causes for eccentricity, including poor manufacturing accuracy and dynamics coupling effects [43]. The eccentricity $l_g$ can be decomposed into $x$- and $z$-axes, and the resulting components are expressed as $\varepsilon_x$ and $\varepsilon_z$. In this study, the dynamic eccentricity in the $z$ direction due to the road excitation of the vehicle while driving is mainly investigated. the UEMF decomposition in the vertical direction is shown in Figure 2.

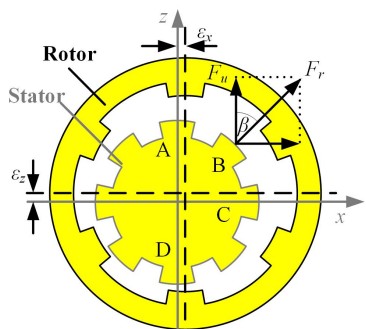

**Figure 2.** The vertical UEMF induced by eccentricity.

Based on the UEMF definition and the mixed-eccentricity, the vertical UEMF $F_u$ is [11]:

$$F_u = \sum_{k=1}^{4} \left[ \left( -\frac{T_k \delta}{l_g - \varepsilon_y \cos \beta_k} + \frac{T_k \delta}{l_g + \varepsilon_y \cos \beta_k} \right) \cos \beta_k \right] \tag{9}$$

where $\beta$ is the phase structure angle ($\beta_1 = 0°, \beta_2 = 45°, \beta_3 = 90°$, and $\beta_4 = 135°$) and the nominal air gap is 0.8 mm.

The strategy uses a current chopping controller to avoid the stator winding resistance stemming from consuming the electrical energy; the preset current limit is 23 A. Due to the configuration of the adopted SRM model, the given turn-on angle $\theta_{\text{on}}$ ranges between $0°$ and $12°$, while the turn-off angle $\theta_{\text{off}}$ range is between $22°$ and $30°$. In this paper, the main role of SRM is to provide driving torque. The driving torque mean value $T_{\text{mean}}$ and standard deviation $T_{\text{std}}$ were used as indexes, and the results are shown in Figure 3a,b. Based on the results, angle $\theta_{\text{on}}$ and $\theta_{\text{off}}$ values are selected as $4°$ and $28°$, respectively. Using these control parameters, the single-phase UEMF of SRM in both vertical and longitudinal directions are characterized by different $\varepsilon_x$ and $\varepsilon_z$ values. The results are shown in Figure 3c.

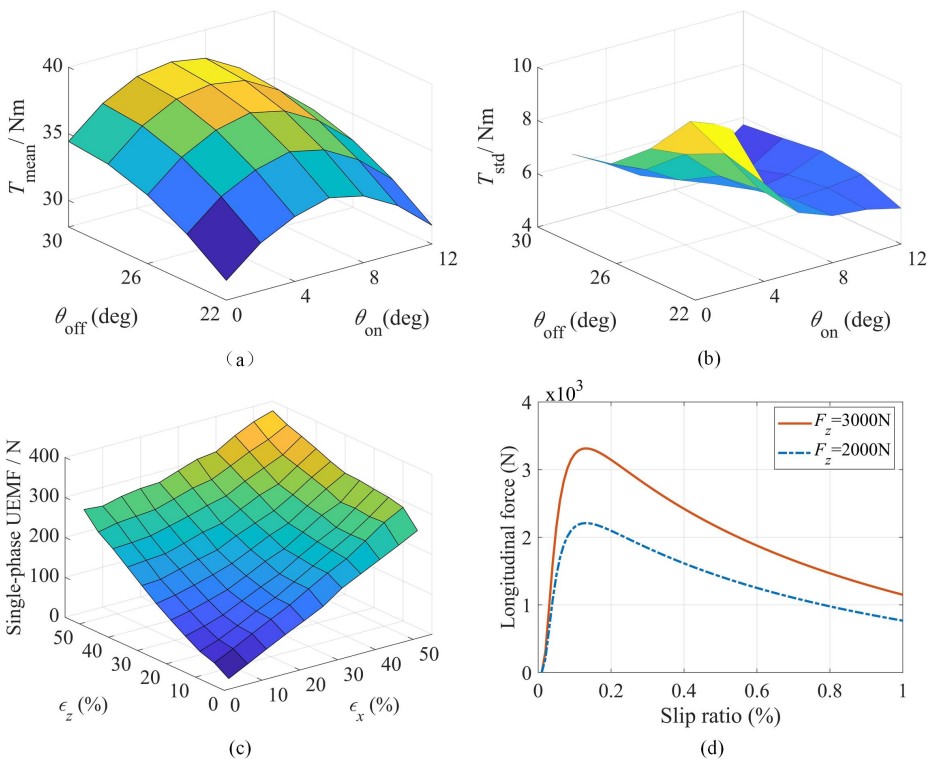

**Figure 3.** The IWM characteristics: (**a**) $T_{\text{mean}}$ versus turning angle; (**b**) $T_{\text{std}}$ versus turning angle; (**c**) single-phase UEMF of SRM; (**d**) the magic formula.

To simplify the calculation, it was assumed that the motor stator eccentricity direction of is vertical. The eccentricity can directly affect the UEMF, which is represented in form of the electromagnetic coupling of the system. The wheel rotation equilibrium equation is as follows:

$$\begin{cases} I_t \dot{\omega}_t = T - F_x R_t - M_r \\ M_r = \mu R_t F_z \\ F_z = \sum mg - F_{tz} \end{cases} \tag{10}$$

where $I_t$ is the rotational inertia of the total wheel, $R_t$ is the effective rolling radius, $M_r$ is the rolling resistance moment generated by the tire, $F_x$ is the reaction force between the tire and the road obtained by the magic formula [35] (see Figure 3d), $T$ is the motor drive torque, $F_z$ is the vertical dynamic load, $\mu$ is rolling resistance coefficient, $\sum m$ is the quarter total vehicle mass, and $F_{tz}$ is vertical tire force discussed in Section 2.2.

### 2.2. The FCT Model

Next, the FCT model, which applies both the RRM and FRM, was integrated to capture the transient tire-road contact patch and tire belt deformation.

The road roughness $z_r$ is commonly described via power spectral density in the vertical direction. The Harmonic superposition algorithm was used to generate time-domain road profiles [44,45]:

$$z_r(t) = \sum_{K=1}^{M} \sqrt{2 \cdot G_q(f_{\text{mid}-K}) \cdot \frac{f_2 - f_1}{M}} \sin(2\pi f_{\text{mid}-K} t + \phi_K) \tag{11}$$

where $f_{\text{mid}-K}$ is the $K$-th middle frequency and $G_q(f_{\text{mid}-K})$ is the power spectral density at $f_{\text{mid}-K}$. Further, $\Phi_K$ is an identifiably distributed phase with the range of $(0, 2\pi)$. The upper and lower time-domain frequency boundaries are denoted as $f_1$ and $f_2$. Finally, according to the ISO-8608 [46], ISO-A, ISO-B, and ISO-C road displacement spectrum as shown in Figure 4. In this paper, the ISO-B was adopted as the actual road excitation.

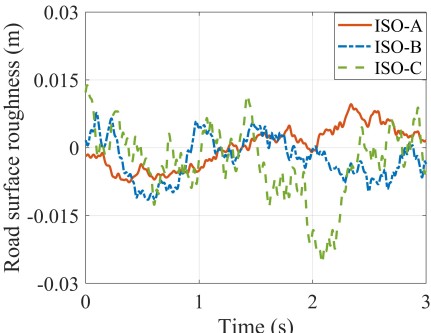

**Figure 4.** The road displacement spectrums.

When a tire is loaded on the road, a large deformation having a limited contact length occurs near the contact patch, which could be equated to a contact point [47]. Moreover, the vertical residual stiffness of RRM was introduced to determine the overall vertical tire stiffness [36]. The residual stiffness $k_{rs}$ is equal to:

$$k_{rs} = z_r - z_t + q_{V1}\omega_t{}^2 \tag{12}$$

where $z_t$ is the vertical tire displacement, $\omega_t$ is the angular tire velocity, $q_{V1}$ is the vertical stiffness correlation coefficient of the tire.

The vertical contact point force $F_{tzc}$ is directly related to the $k_{rs}$. The tire deformation within the contact patch can be equated to the contact point deformation using a deformation stiffness of $k_{tr}$, as shown in Figure 5. After neglecting the higher-order terms, a third-order polynomial was used to describe the vertical force due to the residual tire deflection [36]:

$$F_{tzc} = q_{Fzr3}k_{rs}^3 + q_{Fzr2}k_{rs}^2 + q_{Fzr1}k_{rs} \tag{13}$$

where $q_{Fzr*}$ are polynomial coefficients expressed as:

$$\begin{cases} q_{Fzr1} = k_t \frac{q_{Fz1}(1+q_{V2}|\omega_t|)}{k_t - q_{Fz1}(1+q_{V2}|\omega_t|)} \\ q_{Fzr2} = k_t \frac{k_t(k_t \cdot q_{Fz2} + q_{Fzr1} \cdot q_{Fz2})(1+q_{V2}|\omega_t|)}{(k_t - q_{Fz1}(1+q_{V2}|\omega_t|))^2} \\ q_{Fzr3} = 2k_t \frac{q_{Fzr2} \cdot q_{Fz2}(1+q_{V2}|\omega_t|)}{(k_t - q_{Fz1}(1+q_{V2}|\omega_t|))^2} \end{cases} \tag{14}$$

where $q_{V2}$, $q_{Fz1}$, and $q_{Fz2}$ are vertical stiffness correlation coefficients of tire and $k_t$ is its sidewall stiffness.

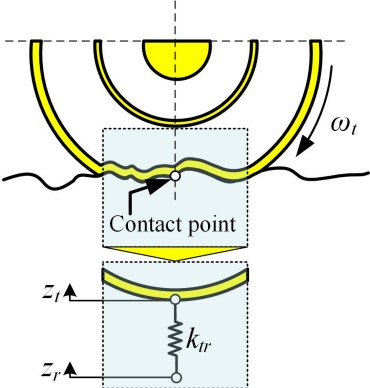

**Figure 5.** The equivalent process for deformation.

The RRM only considers the contact point deformation; the tire force generated by the tire belt in the contact patch is not calculated—the FRM was used to calculate it. The model includes a finite number of independent radial springs and damping elements evenly

distributed in the lower semicircle, as shown in Figure 6 [48]. The total number of the discrete radial elements is denoted as $N_{tr}$ [49].

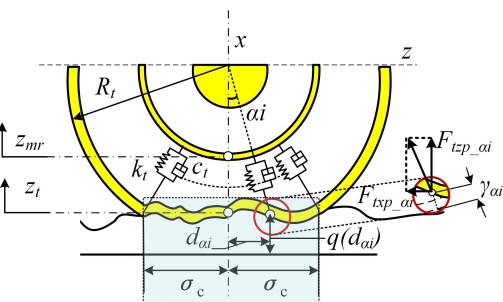

**Figure 6.** The illustrative representation of the FRM.

In Figure 6, $\sigma_c$ is equal to the half of the contact patch, $z_{mr}$ is the vertical coordinate of the rotor center, $z_t$ is the vertical contact point displacement, $c_t$ denotes the tire damping, and $\alpha i$ represents the angle between the arbitrarily chosen element and the vertical, ranging from 1 to $N_{tr}$.

For the positive $x$-axis, the subscript $i$ indicates the element sequence number ranging from $-\arcsin(\sigma_c/R_t)$ to $+\arcsin(\sigma_c/R_t)$. Further, $\gamma_{\alpha i}$, $d_{\alpha i}$, and $q(d_{\alpha i})$ are the radial deformation, driving distance, and the element road elevation, respectively. As a result of tire deformation, the vertical component $F_{tzp\_\alpha i}$ of the radial spring and damping element forces are:

$$F_{tzp-\alpha i} = \begin{cases} (\gamma_{\alpha i}k_{trd} + \dot{\gamma}_{\alpha i}c_{trd})\cos\alpha i & \gamma_{\alpha i} > 0 \\ \dot{\gamma}_{\alpha i}c_{trd}\cos\alpha i & \gamma_{\alpha i} \leq 0 \end{cases} \tag{15}$$

The deformation and deformation rate of a certain radial element can be approximately calculated using:

$$\begin{cases} \alpha i = -\arcsin(\sigma_c/R_t) + 2i\arcsin(\sigma_c/R_t)/N_{tr} \\ d_{\alpha i} = x_t + \sqrt{(R_t^2 - \sigma_c^2)} \cdot \tan\alpha i \\ (R_t - \gamma_{\alpha i})\cos\alpha i + q(dis_{\alpha i}) = R_t + z_{mr} + z_t \end{cases} \tag{16}$$

where $x_t$ is the longitudinal displacement of the tire. The vertical tire force component $F_{tzp}$ caused by tire deformation at the rotor center can be obtained by summing up the force components of each spring and damping element:

$$F_{tzp} = \sum_{i=1}^{N_{tr}} F_{txp-\alpha i}. \tag{17}$$

In general, the vertical tire force calculated through the FTC model under road roughness excitation is:

$$F_{tz} = F_{tzc} + F_{tzp} \tag{18}$$

Finally, $F_{tz}$ is further employed as an external excitation to the IVES (instead of the road roughness).

### 2.3. DVAS-ASS Integrated Model

The quarter models of three types of suspension structures were created in this section. Since the sprung mass distribution coefficients in most modern vehicles are designed to be close to 1. Furthermore, in the suspension structure of the IWMD EVs configured with ASS, the four wheels are independent of each other in vertical motion. Therefore, the quarter model is representative when studying the vertical motion of the vehicle [50]. The sprung and unsprung mass are included, as well as the suspension and tires.

Figure 7a shows a passive-suspension system in which the IWM stator and rotor are rigidly connected to the axle and hub, respectively. Therefore, the IWM becomes the

unsprung vehicle mass; $m_s$ is the quarter of the vehicle sprung mass, $m_{mr}$ is the rotor mass, $m_{ms}$ is the sum of the stator, axle, and tire mass, $k_b$ is the motor bearing stiffness, $k_s$ is the suspension stiffness, $c_s$ is the suspension damping, $F_u$ is the UEMF, and $z_*$ represents displacement (where index "$*$" stands for $r$, $mr$, $ms$, $s$, or $ax$). In a passive-suspension system, the IWM is directly impacted due to rigid joints, which deteriorates its reliability and service life. On the other hand, the IWM equipped with the DVAS is a good solution to suppress the vibration problem, as shown in Figure 7b. Variable $m_{ax}$ represents the sum of the axle and tire masses, while $k_d$ and $c_d$ are the stiffness and damping of the absorber, respectively.

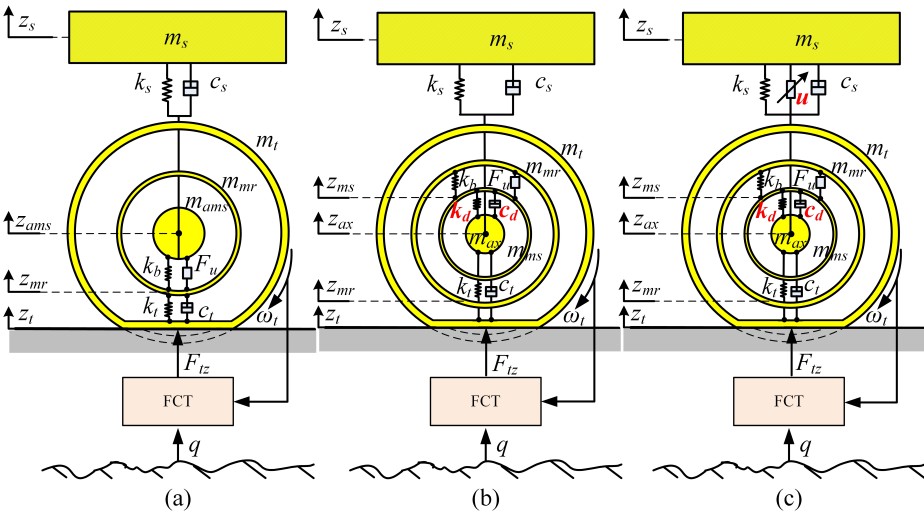

**Figure 7.** Quarter vehicle models equipped with the IWM: (**a**) passive suspension, (**b**) DVAS, (**c**) ASS and DVAS.

The DVAS mechanism is, in theory, similar to that found in passive suspensions. To simultaneously improve the vibration performance of both the IWM and the sprung mass, a suspension combining the ASS and the DVAS was used as the object; its configuration is shown in Figure 7c, $u$ is the active-controlled force of the ASS.

The dynamics of models shown in Figure 7a–c can be described by Newton's motion law, as follows:

$$\begin{cases} m_s\ddot{z}_s + k_s(z_s - z_{ms}) + c_s(\dot{z}_s - \dot{z}_{ms}) = 0 \\ m_{ams}\ddot{z}_{ams} + k_s(z_{ams} - z_s) + c_s(\dot{z}_{ams} - \dot{z}_s) + k_b(z_{ams} - z_{mr}) + F_u = 0 \\ m_{mr}\ddot{z}_{mr} + k_t(z_{mr} - z_t) + c_t(\dot{z}_{mr} - \dot{z}_t) + k_b(z_{mr} - z_{ams}) - F_u = 0 \\ m_t\ddot{z}_t + k_t(z_t - z_{mr}) + c_t(\dot{z}_t - \dot{z}_{mr}) + F_{tz} = 0 \end{cases} \quad (19)$$

$$\begin{cases} m_s\ddot{z}_s + k_s(z_s - z_{ax}) + c_s(\dot{z}_s - \dot{z}_{ax}) = 0 \\ m_{mr}\ddot{z}_{mr} + k_b(z_{mr} - z_{ms}) + F_u = 0 \\ m_{ms}\ddot{z}_{ms} + k_d(z_{ms} - z_{ax}) + c_d(\dot{z}_{ms} - \dot{z}_{ax}) + k_b(z_{ms} - z_{mr}) - F_u = 0 \\ m_{ax}\ddot{z}_{ax} + k_d(z_{ax} - z_{ms}) + c_d(\dot{z}_{ax} - \dot{z}_{ms}) + \\ + k_t(z_{ax} - z_t) + c_t(\dot{z}_{ax} - \dot{z}_t) + k_s(z_{ax} - z_s) + c_s(\dot{z}_{ax} - \dot{z}_s) = 0 \\ m_t\ddot{z}_t + k_t(z_t - z_{ax}) + c_t(\dot{z}_t - \dot{z}_{ax}) + F_{tz} = 0 \end{cases} \quad (20)$$

$$\begin{cases} m_s\ddot{z}_s + k_s(z_s - z_{ax}) + c_s(\dot{z}_s - \dot{z}_{ax}) + u = 0 \\ m_{mr}\ddot{z}_{mr} + k_b(z_{mr} - z_{ms}) + F_u = 0 \\ m_{ms}\ddot{z}_{ms} + k_d(z_{ms} - z_{ax}) + c_d(\dot{z}_{ms} - \dot{z}_{ax}) + k_b(z_{ms} - z_{mr}) - F_u = 0 \\ m_{ax}\ddot{z}_{ax} + k_d(z_{ax} - z_{ms}) + c_d(\dot{z}_{ax} - \dot{z}_{ms}) + \\ + k_t(z_{ax} - z_t) + c_t(\dot{z}_{ax} - \dot{z}_t) + k_s(z_{ax} - z_s) + c_s(\dot{z}_{ax} - \dot{z}_s) - u = 0 \\ m_t\ddot{z}_t + k_t(z_t - z_{ax}) + c_t(\dot{z}_t - \dot{z}_{ax}) + F_{tz} = 0 \end{cases} \quad (21)$$

By combining the models shown above, the overall logical diagram of the proposed mechanical-electrical-magnetic coupling model of the IVES was obtained and is shown

in Figure 8. For a given road profile (ISO-B was used in this paper), the vehicle speed determines the vertical road excitation $z_r$. The road excitation causes wheel vibration, which results in the eccentricity between the rotor and stator. The coupling effect between $\varepsilon_z$ and the UEMF model produces $F_u$. For the driving model, $z_r$, $z_t$, and wheel speed in unison determine the $F_{tz}$ and $F_z$. Further, $F_z$ and the slip ratio between vehicle and wheel determine the magnitude of $F_x$, while the current chopping controller adjusts the voltage (on/off) of the IWM. The control signal of the IWM is adjusted according to the difference between the vehicle and the set speed, which is the input to the vibration model and $w_t$, closing the loop.

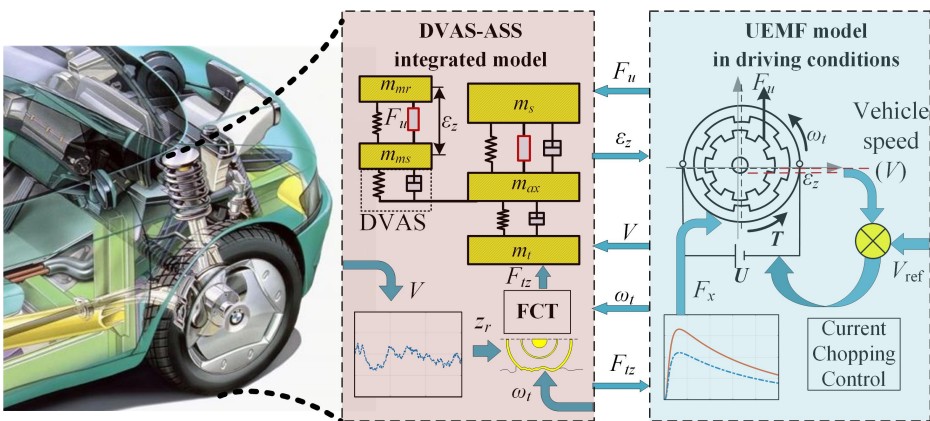

**Figure 8.** The Multi-field coupling model of IVES.

## 3. The IVES Optimization Control

In this section, the UEMF effects on the suspension system were analyzed, and the outcomes were improved by developing a delay-dependent $H\infty$ controller for the ASS.

### 3.1. The UEMF Influence on the Dynamic Performances of the Vehicle

Based on the above-mentioned mathematical models, a simulation platform for the vehicle multi-field coupling dynamics can be developed using MatLab/Simulink. Test vehicle specifications are listed in Table 1. The DVAS "tire" type [11] and air spring active suspension [51] were selected as actuators in this paper. The tire used in this paper is a passenger car summer tire designated as 205/55R16 [52].

**Table 1.** Test vehicle specifications.

| Parameters | Value | Parameters | Value |
|---|---|---|---|
| **DVAS-ASS Integrated Model** | | **Passive Suspension System** | |
| $k_s$ | $3.2 \times 10^4$ N/m | $m_{ams}$ | 34.5 kg |
| $c_s$ | $1.8 \times 10^3$ N·s/m | Driving conditions | |
| $k_b$ | $2.08 \times 10^7$ N/m | $R_t$ | 0.3160 m |
| $k_d$ | $5.3 \times 10^4$ N/m | $\mu$ | 0.0066 |
| $c_d$ | $1.9 \times 10^3$ N·s/m | $I_t$ | 0.546 kg·m$^2$ |
| $k_t$ | $1.8 \times 10^6$ N/m | FTC model | |
| $c_t$ | $510 \times$ N·s/m | $q_{V1}$ | $8.5352 \times 10^{-8}$ m s$^2$ |
| $m_s$ | 332 kg | $q_{V2}$ | $8.81 \times 10^4$ s |
| $m_{ax}$ | 25 kg | $q_{Fz1}$ | $1.4389 \times 10^5$ N/m |
| $m_{ms}$ | 9.5 kg | $q_{Fz2}$ | $4.5090 \times 10^6$ N/m$^2$ |
| $m_{mr}$ | 22.5 kg | | |
| $m_t$ | 6.15 kg | | |
| $\Sigma m$ | 389 kg | | |

The dynamic coupling relationships were investigated assuming the ISO-B road at a speed of 40 m/s. The drive lasted for 3 s and the Fast Fourier Transform (FFT) was used to derive spectral features. For observation, 1–2 s data points were selected for further statistical analysis. The responses of sprung mass acceleration (SMA) and eccentricity (ECC) with and without the UEMF under the pavement excitation are shown in Figure 9.

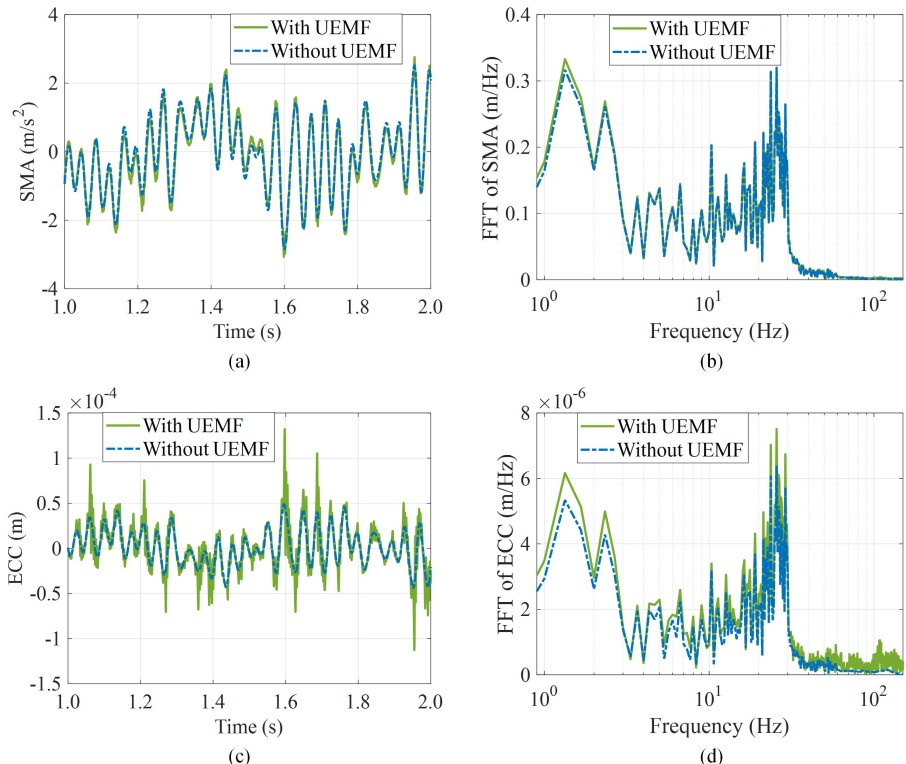

**Figure 9.** Coupling effects between the SMA, the ECC, and the UEMF: (**a**) the SMA time response; (**b**) the SMA frequency response; (**c**) the ECC time response; (**d**) the ECC frequency response.

The time-response results have shown that the SMA and ECC are larger when considering the UEMF effects. The same trend can be deduced from the frequency response—the effect of UEMF on the ECC at high frequencies is particularly pronounced.

The eccentricity between the stator and rotor was further exacerbated by the UEMF generated by the multi-field coupling effect. Such eccentricity, in turn, furthers the UEMF, which is complementary to it. This vicious cycle exacerbates motor vibration, shortening the IWM service life and reducing vehicle comfort. Therefore, it is worthwhile to investigate the issue of how to restrict the multi-field coupling effects.

### 3.2. Active Suspension Control

An $H\infty$ control scheme with strong robustness was used to design the active suspension controllers. The controllers were needed to ensure excellent control effect of the suspension system when given complicated electromagnetic force excitation and unmodeled dynamic perturbation. According to Equation (21), the following set of state variables was selected:

$$
\begin{aligned}
\mathbf{x}(t) = & \left[\dot{z}_s(t) \quad \dot{z}_{mr}(t) \quad \dot{z}_{ms}(t) \quad \dot{z}_{ax}(t) \quad \dot{z}_t(t)\ldots \right. \\
& \left. z_{ax}(t) - z_s(t) \quad z_{ms}(t) - z_{mr}(t) \quad z_{ax}(t) - z_{ms}(t) \quad z_{ax}(t) - z_t(t)^T\right]
\end{aligned}
\tag{22}
$$

The dynamic equations of the ASS with the DVAS system can be written using the following state-space form:

$$\dot{\mathbf{x}}(t) = \mathbf{A}\mathbf{x}(t) + \mathbf{B}_w\mathbf{w}(t) + \mathbf{B}_u\mathbf{u}(t) \tag{23}$$

where:

$$\mathbf{A} = \begin{bmatrix} -c_s m_s^{-1} & 0 & 0 & c_s m_s^{-1} & 0 & k_s m_s^{-1} & 0 & 0 & 0 \\ 0 & 0 & 0 & 0 & 0 & 0 & k_b m_{mr}^{-1} & 0 & 0 \\ 0 & 0 & -c_d m_{ms}^{-1} & c_d m_{ms}^{-1} & 0 & 0 & -k_b m_{ms}^{-1} & k_d m_{ms}^{-1} & 0 \\ c_s m_{ax}^{-1} & 0 & c_d m_{ax}^{-1} & -(c_d + c_s + c_t)m_{ax}^{-1} & c_t m_{ax}^{-1} & -k_s m_{ax}^{-1} & 0 & -k_d m_{ax}^{-1} & -k_t m_{ax}^{-1} \\ 0 & 0 & 0 & c_t m_t^{-1} & c_t m_t^{-1} & 0 & 0 & 0 & k_t m_t^{-1} \\ -1 & 0 & 0 & 1 & 0 & 0 & 0 & 0 & 0 \\ 0 & -1 & 1 & 0 & 0 & 0 & 0 & 0 & 0 \\ 0 & 0 & -1 & 1 & 0 & 0 & 0 & 0 & 0 \\ 0 & 0 & 0 & 1 & -1 & 0 & 0 & 0 & 0 \end{bmatrix}$$

$$\mathbf{B}_w = \begin{bmatrix} 0 & 0 & 0 & 0 & -m_t^{-1} & 0 & 0 & 0 & 0 \\ 0 & -m_{mr}^{-1} & m_{ms}^{-1} & 0 & 0 & 0 & 0 & 0 & 0 \end{bmatrix}^T$$

$$\mathbf{B}_u = \begin{bmatrix} -m_s^{-1} & 0 & 0 & m_{ax}^{-1} & 0 & 0 & 0 & 0 & 0 \end{bmatrix}^T$$

$$\mathbf{u}(t) = \begin{bmatrix} u(t) \end{bmatrix} \quad \mathbf{w} = \begin{bmatrix} F_{tz} & F_u \end{bmatrix}^T$$

Generally, the suspension performance is evaluated based on key factors such as ride comfort, suspension working space, and road-holding ability. Additionally, as for the IWMD EVs, the IWM UEMF is an essential performance factor. Thus, four system responses are investigated: SMA: $\ddot{z}_s$, rattle space (RS): $(z_s - z_{ax})$, tire deformation (TD): $(z_{ax} - z_t)$, and ECC: $(z_{ms} - z_{mr})$. Among these, the first three are used for evaluating the suspension performance, whereas the last one reflects the IWM vibration level.

Focusing on ride comfort improvement and the IWM working environment, the SMA and eccentricity should be minimized, while the other two factors are strict constraints that must be satisfied. Therefore, minimizing the disturbance transfer function norm ($F_u$ and $F_{tz}$) to the control output ($\ddot{z}_s$ and $(z_{ms} - z_{mr})$) is our main goal. The constraints as follows:

$$\begin{aligned} |z_{ax} - z_s| &\leq z_{1\max} \\ |z_{ax} - z_t| &\leq z_{2\max} \\ |u(t)| &\leq u_{\max} \end{aligned} \tag{24}$$

where $z_{1max}$ and $z_{2max}$ are the maximum suspension and tire deflection, respectively. $u_{max}$ is the maximum possible actuator output force.

Based on the above-presented conditions, the performance control output and constrained control output are defined next:

$$\mathbf{z}_1 = \begin{bmatrix} \ddot{z}_s & z_{ms} - z_{mr} \end{bmatrix}^T \tag{25}$$

$$\mathbf{z}_2 = \begin{bmatrix} \dfrac{z_{ax}(t) - z_s(t)}{z_{1\max}} & \dfrac{z_{ax}(t) - z_t(t)}{z_{2\max}} & \dfrac{u(t)}{u_{\max}} \end{bmatrix}^T \tag{26}$$

The state-space equations of the ASS and DVAS for the IWMD EVs can be described using:

$$\begin{aligned} \dot{\mathbf{x}}(t) &= \mathbf{A}\mathbf{x}(t) + \mathbf{B}_w\mathbf{w}(t) + \mathbf{B}_u\mathbf{u}(t) \\ \mathbf{z}_1 &= \mathbf{C}_1\mathbf{x}(t) + \mathbf{D}_1\mathbf{u}(t) \\ \mathbf{z}_2 &= \mathbf{C}_2\mathbf{x}(t) + \mathbf{D}_2\mathbf{u}(t) \end{aligned} \tag{27}$$

where:

$$\mathbf{C}_1 = \begin{bmatrix} -c_s m_s^{-1} & 0 & 0 & c_s m_s^{-1} & 0 & k_s m_s^{-1} & 0 & 0 & 0 \\ 0 & 0 & 0 & 0 & 0 & 0 & 1 & 0 & 0 \end{bmatrix}$$

$$\mathbf{D}_1 = \begin{bmatrix} -m_s^{-1} & 0 \end{bmatrix}^T$$

$$\mathbf{C}_2 = \begin{bmatrix} 0 & 0 & 0 & 0 & 0 & z_{1\,\mathrm{max}}^{-1} & 0 & 0 & 0 \\ 0 & 0 & 0 & 0 & 0 & 0 & 0 & 0 & z_{2\,\mathrm{max}}^{-1} \\ 0 & 0 & 0 & 0 & 0 & 0 & 0 & 0 & 0 \end{bmatrix}$$

$$\mathbf{D}_2 = \begin{bmatrix} 0 & 0 & u_{\mathrm{max}}^{-1} \end{bmatrix}^T$$

The ASS inevitably has operating time delays, with the majority being the response time delay of the actuator. Hence, time delay should not be neglected when designing the active suspension system.

Assuming that the time-varying delay $\tau(t)$ in a closed-loop controlled suspension system satisfies the condition:

$$0 < \tau(t) < \tau_{\mathrm{max}} \tag{28}$$

where $\tau_{max}$ represents the upper bound of the delay time. Considering the delay time, the output feedback controller can be described as:

$$\mathbf{u}(t) = \mathbf{u}(t - \tau) = \mathbf{K}_t \mathbf{x}(t - \tau) \tag{29}$$

where $K_t$ is the output controller gain. Next, the ASS that considers the time delay in the control system can be expressed via:

$$\begin{aligned} \dot{\mathbf{x}}(t) &= \mathbf{A}\mathbf{x}(t) + \mathbf{B}_w \mathbf{w}(t) + \mathbf{B}_u \mathbf{K}_t \mathbf{x}(t - \tau) \\ \mathbf{z}_1 &= \mathbf{C}_1 \mathbf{x}(t) + \mathbf{D}_1 \mathbf{K}_t \mathbf{x}(t - \tau) \\ \mathbf{z}_2 &= \mathbf{C}_2 \mathbf{x}(t) + \mathbf{D}_2 \mathbf{K}_t \mathbf{x}(t - \tau) \end{aligned} \tag{30}$$

In this paper, the aim is to design a state feedback controller $\mathbf{u}(t) = \mathbf{K}_t \mathbf{x}(t - \tau)$ that will meet the following conditions:

(1) Without external perturbations, the closed-loop system shown in Equation (30) is asymptotically stable.

(2) The performance $\|\mathbf{z_1}(t)\|_\infty \le \gamma w(t)_\infty$ is minimized subject to Equation (30), where $\gamma$ is the bounded $H\infty$ parametrization.

(3) The time-domain constraint $|\mathbf{z_2}(t)| \le 1$ must be satisfied.

Considering the time delay, the $H\infty$ controller was designed through the following steps:

Given positive scalars $\gamma > 0$, $\eta > 0$, and $\rho > 0$, for any time delay $t$ satisfying $0 < \tau(t) < \tau_{\mathrm{max}}$, the system shown in Equation (26) with the controller from Equation (29) is asymptotically stable with $w(t) = 0$. In that case, it also satisfies the performance described in Equations (25) and (26) for $w(t) \in [0, \infty)$, given that symmetric matrices $\mathbf{P} > 0$, $\mathbf{Q} > 0$, and $\mathbf{R} > 0$ exist, and a general matrix $\mathbf{K}_t$ satisfying the following linear matrix equations (LMIs) is:

$$\begin{bmatrix} \mathbf{\Omega} & \mathbf{PB}_u\mathbf{K}_t & \mathbf{0} & \mathbf{PB}_w & \sqrt{\tau_{\mathrm{max}}}\mathbf{A}^T\mathbf{P} & \mathbf{C}_1^T \\ * & -\tau_{\mathrm{max}}^{-1}\mathbf{Q} & \mathbf{0} & \mathbf{0} & \sqrt{\tau_{\mathrm{max}}}(\mathbf{B}_u\mathbf{K}_t)^T\mathbf{P} & (\mathbf{D}_1\mathbf{K}_t)^T \\ * & * & -\mathbf{R} & \mathbf{0} & \mathbf{0} & \mathbf{0} \\ * & * & * & -\gamma^2\mathbf{I} & \sqrt{\tau_{\mathrm{max}}}\mathbf{B}_w^T\mathbf{P} & \mathbf{0} \\ * & * & * & * & -2\eta\mathbf{P} + \eta^2\mathbf{Q} & \mathbf{0} \\ * & * & * & * & * & -\mathbf{I} \end{bmatrix} < 0 \tag{31}$$

$$\begin{bmatrix} -\mathbf{P} & (\mathbf{C}_2 + \mathbf{D}_2\mathbf{K}_t)^T \\ * & -\frac{1}{\rho}\mathbf{I} \end{bmatrix} < 0 \tag{32}$$

where: $\mathbf{\Omega} = sys[\mathbf{P}(\mathbf{A} + \mathbf{B}_w\mathbf{K}_t)] + \mathbf{R}$

For comparison, the $H\infty$ controller without considering time delay is obtained as follows:

For a given scalar $\rho_c > 0$, if $\gamma_c > 0$ and a positive definite symmetric matrix $\mathbf{X}$ exists, making the following inequalities feasible under LMIs:

$$\begin{bmatrix} (\mathbf{AX} + \mathbf{B_2W})^T + (\mathbf{AX} + \mathbf{B_2W}) & \mathbf{B_1} & (\mathbf{C_1X} + \mathbf{D_1W})^T \\ \mathbf{B_1}^T\mathbf{P} & -\gamma_c\mathbf{I} & \mathbf{0} \\ \mathbf{C_1X} + \mathbf{D_1W} & \mathbf{0} & -\gamma_c\mathbf{I} \end{bmatrix} < 0 \tag{33}$$

$$\begin{bmatrix} -\mathbf{X} & \mathbf{C_2X} + \mathbf{D_2W} \\ * & -\frac{1}{\rho_c}\mathbf{I} \end{bmatrix} < 0 \tag{34}$$

The state feedback gain $\mathbf{K}_t$ can be given by $\mathbf{K}_t = \mathbf{WX}^{-1}$

**Proof.** Lemma 1 [53]: For any matrix $\mathbf{X} > 0$, matrices (or scalars) $\mathbf{M}$ and $\mathbf{N}$ with compatible dimensions, the following inequality holds:

$$-2\mathbf{M}^T\mathbf{N} \le \mathbf{M}^T\mathbf{X}^{-1}\mathbf{M} + \mathbf{N}^T\mathbf{XN} \tag{35}$$

Since $x(t) - x(t - \tau) - \int_{t-\tau(t)}^t \dot{x}(\alpha)d\alpha = 0$, by adding it to Equation (30):

$$\dot{\mathbf{x}}(t) = (\mathbf{A} + \mathbf{B_2K})\mathbf{x}(t) + \mathbf{B_1w}(t) - \mathbf{B_2K}_t \int_{t-\tau(t)}^t \dot{x}(t)\mathrm{d}t \tag{36}$$

Selecting the Lyapunov function shown below to analyze the system stability we obtain:

$$\mathbf{V}(t) = \mathbf{V}_1(t) + \mathbf{V}_2(t) + \mathbf{V}_3(t) \tag{37}$$

where:

$$\mathbf{V}_1(t) = \mathbf{x}^T(t)\mathbf{Px}(t)\quad \mathbf{V}_2(t) = \int_{t-\tau_{\max}}^t \mathbf{x}^T(s)\mathbf{Rx}(s)d_s\quad \mathbf{V}_3(t) = \int_{-\tau_{\max}}^0 \int_{t+\theta}^t \mathbf{x}^T(s)\mathbf{Qx}(s)d_sd_\theta \tag{38}$$

with $\mathbf{P}$, $\mathbf{Q}$, and $\mathbf{R}$ being symmetric positive definite matrices. The derivative of $\mathbf{V}_1(t)$ is shown next:

$$\begin{aligned} \dot{\mathbf{V}}_1(t) =\ & \mathbf{x}^T(t)\left\{\mathbf{P}[\mathbf{A} + \mathbf{B_2K}_t] + [\mathbf{A} + \mathbf{B_2K}_t]^T\mathbf{P}\right\}\mathbf{x}(t) \\ & + \mathbf{x}^T(t)\mathbf{PB_1w}(t) + \mathbf{w}^T(t)\mathbf{B}_1^T\mathbf{Px}(t) - 2\mathbf{x}^T(t)\mathbf{PB_2K}_t\int_{t-\tau_{\max}}^t \dot{x}(t)\mathrm{d}t \end{aligned} \tag{39}$$

Using **Lemma 1**, that gets:

$$\begin{aligned} \dot{\mathbf{V}}_1(t) \le\ & \mathbf{x}^T(t)\left\{\mathbf{P}[\mathbf{A} + \mathbf{B_2K}_t] + [\mathbf{A} + \mathbf{B_2K}_t]^T\mathbf{P}\right\}\mathbf{x}(t) \\ & + \mathbf{x}^T(t)\mathbf{PB_1w}(t) + \mathbf{w}^T(t)\mathbf{B}_1^T\mathbf{Px}(t) + \\ & \tau_{\max}\mathbf{x}^T(t)\mathbf{PB_2K_t}\mathbf{Q}^{-1}(\mathbf{PB_2K}_t)^T\mathbf{x}(t) + \int_{t-\tau_{\max}}^t \dot{x}(t)^T\mathbf{Q}\dot{x}(t)\mathrm{d}t \end{aligned} \tag{40}$$

Similarly, derivatives of $\mathbf{V}_2(t)$ and $\mathbf{V}_3(t)$ are expressed as:

$$\begin{aligned} \dot{\mathbf{V}}_2(t) &= \mathbf{x}^T(t)\mathbf{Rx}(t) - \mathbf{x}^T(t - \tau_{\max})\mathbf{Rx}(t - \tau_{\max}) \\ \dot{\mathbf{V}}_3(t) &= \tau\mathbf{x}^T(t)\mathbf{Q}\dot{x}(t) - \int_{t-\tau_{\max}}^t \dot{x}(s)^T\mathbf{Q}\dot{x}(s)\mathrm{d}s \end{aligned} \tag{41}$$

Further, the $\mathbf{V}(t)$ derivative is shown as follows:

$$\begin{aligned} \dot{\mathbf{V}}(t) &= \dot{\mathbf{V}}_1(t) + \dot{\mathbf{V}}_2(t) + \dot{\mathbf{V}}_3(t) \\ &\le \mathbf{x}^T(t)\left\{\mathbf{P}[\mathbf{A} + \mathbf{B_2K}_t] + [\mathbf{A} + \mathbf{B_2K}_t]^T\mathbf{P} + \tau_{\max}(t)\mathbf{PB_2K_t}\mathbf{Q}^{-1}(\mathbf{PB_2K})^T + \mathbf{R}\right\}\mathbf{x}(t) \\ &\quad + \mathbf{x}^T(t)\mathbf{PBP_1w}(t) + \mathbf{w}^T(t)\mathbf{B}_1^T\mathbf{Px}(t) - \mathbf{x}^T(t - \tau_{\max})\mathbf{Rx}(t - \tau_{\max}) + \tau_{\max}\dot{x}(t)^T\mathbf{Q}\dot{x}(t) \end{aligned} \tag{42}$$

Defining $\boldsymbol{\xi}^T(t) = \left[\mathbf{x}^T(t), \mathbf{x}^T(t-\tau), \mathbf{x}^T(t-\tau_{\max}), \mathbf{w}^T(t)\right]$, thus

$$\dot{\mathbf{V}}(t) \leq \boldsymbol{\xi}^T(t)\Xi\boldsymbol{\xi}(t) + \tau_{\max}\dot{\mathbf{x}}(t)^T\mathbf{Q}\dot{\mathbf{x}}(t) \tag{43}$$

where:

$$\Xi = \begin{bmatrix} \boldsymbol{\Theta} & 0 & 0 & \mathbf{PB_1} \\ * & 0 & 0 & 0 \\ * & * & -\mathbf{R} & 0 \\ * & * & * & 0 \end{bmatrix}$$

$\boldsymbol{\Theta} = sys[\mathbf{P}(\mathbf{A} + \mathbf{B_2K_t})] + \mathbf{R} + \tau_{\max}\mathbf{PB_2K_t}\mathbf{Q}^{-1}(\mathbf{PB_2K_t})^T$

The system $H\infty$ performance can be described as:

$$\mathbf{z_1}^T(t)\mathbf{z_1}(t) \leq \gamma^2\mathbf{w}^T(t)\mathbf{w}(t) \tag{44}$$

By adding to Equation (43), it is possible to obtain:

$$\dot{\mathbf{V}}(t) + \mathbf{z}_1^T(t)\mathbf{z}_1(t) - \gamma^2\mathbf{w}^T(t)\mathbf{w}(t)\}\xi^T(t)\boldsymbol{\Pi}_1\xi(t) \tag{45}$$

where:

$$\boldsymbol{\Pi}_1 = \begin{bmatrix} \boldsymbol{\Omega} & \mathbf{PB_2K_t} & 0 & \mathbf{PB_1} & \sqrt{\tau_{\max}}^T\mathbf{A}^T\mathbf{P} & \mathbf{C}_1^T \\ * & -\tau_{\max}^{-1}\mathbf{Q} & 0 & 0 & \sqrt{\tau_{\max}}(\mathbf{B_2K_t})^T\mathbf{P} & (\mathbf{D_1K_t})^T \\ * & * & -\mathbf{R} & 0 & 0 & 0 \\ * & * & * & -\gamma^2\mathbf{I} & \sqrt{\tau_{\max}}\mathbf{B_1}^T\mathbf{P} & 0 \\ * & * & * & * & \mathbf{PR}^{-1}\mathbf{P} & 0 \\ * & * & * & * & * & -\mathbf{I} \end{bmatrix}$$

For any positive scalar $\eta$, the following expression is true:

$$\mathbf{PR}^{-1}\mathbf{P} \leq -2\eta\mathbf{P} + \eta^2\mathbf{R} \tag{46}$$

By replacing the item $\mathbf{PR}^{-1}\mathbf{P}$ in $\prod_1$ with $-2\eta\mathbf{P} + \eta^2\mathbf{R}$, we get $\prod_2$. Moreover, it is true that $\prod_1 \leq \prod_2$, where:

$$\Pi_2 = \begin{bmatrix} \boldsymbol{\Omega} & \mathbf{PB_2K_t} & 0 & \mathbf{PB_1} & \sqrt{\tau_{\max}}\mathbf{A}^T\mathbf{P} & \mathbf{C}_1^T \\ * & -\tau_{\max}^{-1}\mathbf{Q} & 0 & 0 & \sqrt{\tau_{\max}}(\mathbf{B_2K_t})^T\mathbf{P} & (\mathbf{D_1K_t})^T \\ * & * & -\mathbf{R} & 0 & 0 & 0 \\ * & * & * & -\gamma^2\mathbf{I} & \sqrt{\tau_{\max}}\mathbf{B_1}^T\mathbf{P} & 0 \\ * & * & * & * & -2\eta\mathbf{P} + \eta^2\mathbf{Q} & 0 \\ * & * & * & * & * & -\mathbf{I} \end{bmatrix}$$

If the matrix inequality $\prod_2 < 0$, that $\mathbf{z_1}^T(t)\mathbf{z_1}(t) \leq \gamma^2\mathbf{w}^T(t)\mathbf{w}(t)$.

From inequality Equation (45), it can be derived that $\dot{\mathbf{V}}(t) + \mathbf{z}_1^T(t)\mathbf{z}_1(t) - \gamma^2\mathbf{w}^T(t)\mathbf{w}(t) \leq 0$. Integrating both sides of the inequality above from zero to any $t > 0$, the following is obtained:

$$\mathbf{V}(t) - \mathbf{V}(0) + \int_0^t \|\mathbf{z}_1(t)\|_2^2 dt - \gamma^2 \int_0^t \|\mathbf{w}(t)\|_2^2 dt \leq 0 \tag{47}$$

Considering the existence of $\int_0^t \mathbf{w}(t)_2^2 dt \leq w_{\max}$, the above inequality can be further rewritten:

$$\mathbf{V}(t) + \int_0^t \|\mathbf{z}_1(t)\|_2^2 dt \leq \gamma^2 w_{\max} + \mathbf{V}(0) \tag{48}$$

Definition $\gamma^2 w_{\max} + \mathbf{V}(0) = \rho$, where $w_{max}$ is upper perturbation energy bound, can be introduced with a value equal to $w_{\max} = [\rho - \mathbf{V}(0)]/\gamma^2$, Therefore, it can be introduced under $t = 0$ and within a given perturbation suppression regime $\gamma$. In combination with $\int_0^t \|\mathbf{z}_1(t)\|_2^2 dt > 0$, synthetically, the following inequalities hold:

$$\begin{aligned} &\rho\mathbf{x}^T[\mathbf{C}_2 + \mathbf{D_2K_t}]^T[\mathbf{C}_2 + \mathbf{D_2K_t}]\mathbf{x} \\ &= \rho[\mathbf{C_2x} + \mathbf{D_2u}]^T[\mathbf{C_2x} + \mathbf{D_2u}] < x^T\mathbf{P}x < \rho \end{aligned} \tag{49}$$

This causes matrix inequality $-\mathbf{P} + \rho[\mathbf{C}_2 + \mathbf{D}_2\mathbf{K}_t]^T[\mathbf{C}_2 + \mathbf{D}_2\mathbf{K}_t] < 0$. Using the Schur complementary theorem [54], the equation is converted into:

$$\begin{bmatrix} -\mathbf{P} & (\mathbf{C}_2 + \mathbf{D}_2\mathbf{K}_t)^T \\ * & -\frac{1}{\rho}\mathbf{I} \end{bmatrix} < 0 \tag{50}$$

For a case without a time delay controller, the proof can be found in [55].
The proof is completed. □

## 4. Simulation and Analyses

In this section, the influence of the ASS and DVAS on the IWMD EVs performance, as well as the effectiveness of the delay-dependent $H\infty$ active suspension controller is illustrated.

### 4.1. Performance Comparison of Different Structures

Four system responses, the SMA, the RS, the TD, and the ECC were investigated in Section 3. For the maximum time-delay $\tau_{max}$= 40 ms, Equations (31) and (32) are LMIs of variables $\mathbf{P}$, $\mathbf{Q}$, and $\mathbf{R}$. Thus, the program can be solved via the solver-mincx within the LMI toolbox. Through LMI algorithms, the control gain matrix $K_t$ of the ASS controller (considering the time delay) is obtained with the minimum guaranteed closed-loop $H\infty$ performance index $\gamma_t$ = 8.82. Such output implies that for any time-delay satisfying $0 \le \tau \le 40$ ms, the controller can stabilize the system with the $H\infty$ performance.

$$K_t = [-18480.29\ -1392.86\ -587.89\ -1687.74\ -261.17\ 39039.25\ 12710.53\ 6087.22\ -88442.36]$$

Similarly, a conventional $H\infty$ controller without considering the control time delay can be derived. The gain matrix $\mathbf{K}_c$ of the Controller with the minimum guaranteed closed-loop $H\infty$ performance index $\gamma_c$ = 6.28 is obtained as:

$$K_c = [-19081.62\ -1443.61\ -609.36\ -1746.70\ -276.04\ 40139.99\ 13055.35\ 6380.52\ -92866.90]$$

After the method proposed in this study to solve the multi-objective control problem of the ASS, its effectiveness is illustrated through four different cases:

1. Case 1—equipped with the IWM and passive suspension;
2. Case 2—equipped with the DVAS in IWM and passive suspension;
3. Case 3—equipped with the DVAS in IWM and the ASS using the control gain matrix $K_c$;
4. Case 4—equipped with the DVAS in IWM and the ASS using the control gain matrix $K_t$.

Figure 10 shows the controller forces applied on the ASS in cases 3 and 4. Both controller forces are less than umax. Further, the controller force in Case 4 is smaller than that in Case 3.

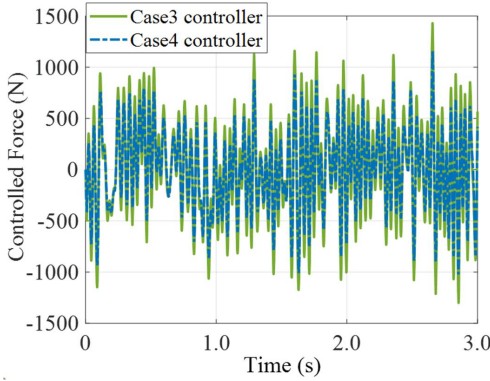

**Figure 10.** Controller forces in Case 3 and Case 4.

The response comparisons of the two-output time and frequency domains of the SMA and ECC are depicted in Figure 11.

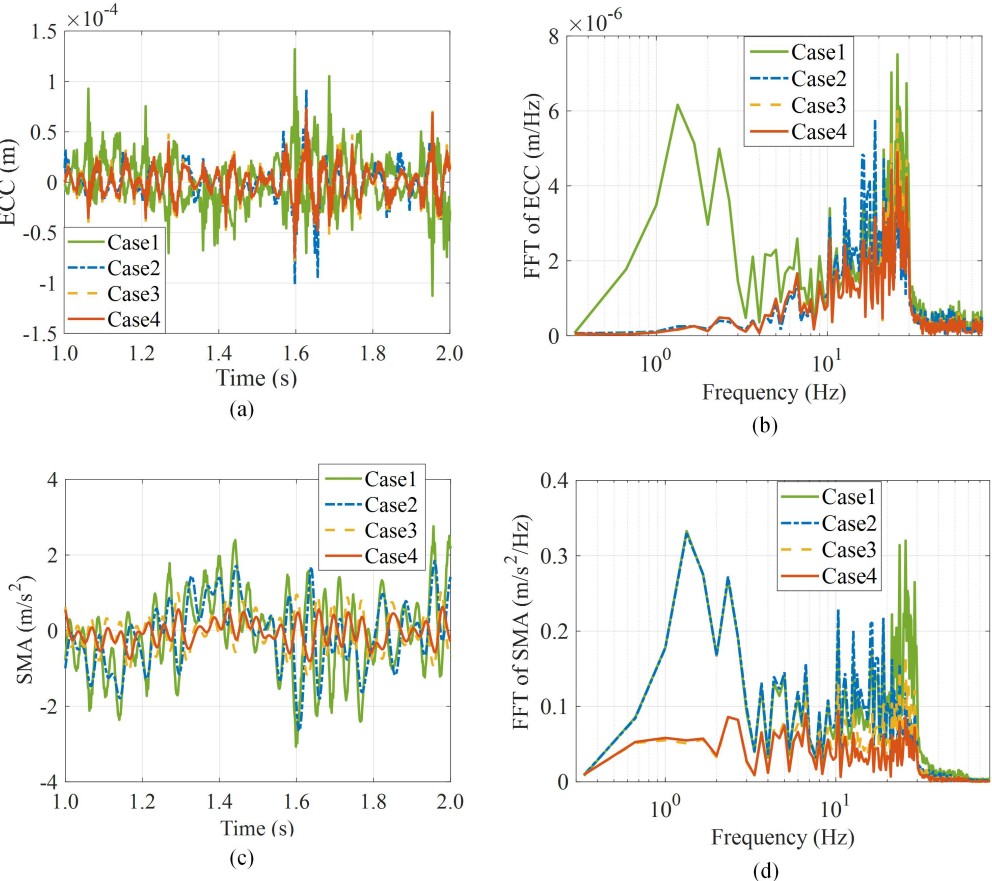

**Figure 11.** Responses of four cases under the road excitation: (**a**) SMA time response; (**b**) SMA frequency response; (**c**) ECC time response; (**d**) ECC frequency response.

The detailed simulation results are available in Figure 11. The responses of the SMA in time and frequency domains are shown in Figure 11a,b, respectively. As shown in Figure 11a, the SMA is significantly smaller in cases 3 and 4 compared to cases 1 and 2, mostly due to the introduction of the ASS. Figure 11b shows that Case 2 can reduce the amplitude of high-frequency compared to Case 1; however, there is no optimization effect at the human-sensitive frequency of 4–8 Hz. Cases 3 and 4 can significantly reduce the amplitude in both high and low frequencies (especially compared to Case 1), especially in the 4–8 Hz range.

The ECC responses in time and frequency domains are shown in Figure 11c,d, respectively. As shown in Figure 11c, in contrast with the SMA optimization effect, all three cases had reduced amplitude in the wide frequency domain concerning Case 1. The proposed control methods of the active suspension are effective for the IWM vibration. The SMA is smaller in cases 3 and 4 compared to in Case 2 in the frequencies between 3 Hz and 12 Hz, indicating that the proposed control method for the ASS can further reduce the impulse force acting on the IWM.

Simulation results for all four cases are provided in Table 2.

**Table 2.** Root Mean Square (RMS) optimization results.

| RMS | SMA (m/s$^2$)-Decrement (%) | ECC ($10^{-5}$ m)-Decrement (%) | RS ($10^{-3}$ m)-Decrement (%) | TD ($10^{-4}$ m)-Decrement (%) |
|---|---|---|---|---|
| Case 1 | 0.9941- | 2.365- | 4.229- | 4.328- |
| Case 2 | 0.8001-↓19.5% | 1.673-↓29.2% | 4.221-↓0.2% | 4.325-↓0.07% |
| Case 3 | 0.4148-↓58.2% | 1.625-↓31.2% | 4.223-↓0.1% | 4.322-↓0.1% |
| Case 4 | 0.3217-↓67.6% | 1.444-↓38.9% | 4.222-↓0.2% | 4.318-↓0.2% |

As can be seen from Table 2, both the DVAS and the ASS can reduce the four-parameter indices, while having different roles in improving comfort and handling. The DVAS has an important role in reducing eccentricity, while the ASS has a decisive effect on spring acceleration. For the RS and TD, the optimization effect is not significant compared to the conventional suspension since they are not optimized in a targeted way. At the same time, the H-infinity algorithm, which considers the time delay, displayed a performance improvement compared to the algorithm not considering it.

Table 3 compares control methods in terms of simulation results considering the presence of time delay in the active suspension actuator. It also indicates that, with the increase in time delay, both Case 3 evaluation indexes deteriorated to different degrees; the SMA deterioration is obvious. In contrast, Case 4 adapted easier and the degree of deterioration was not significant—it was within acceptable limits.

**Table 3.** The RMS optimization results for both cases.

| Case | Time Delay | SMA (m/s$^2$) -Deteriorate | ECC ($10^{-5}$ m) -Deteriorate |
|---|---|---|---|
| Case 3 | $\tau = 0$ ms | 0.4848 | 1.625 |
| | $\tau = 15$ ms | 0.6279-↑29.5% | 1.667-↑2.5% |
| controller | $\tau = 30$ ms | 0.7634-↑57.5% | 1.694-↑4.4% |
| Case 4 | $\tau = 0$ ms | 0.3217 | 1.444 |
| | $\tau = 15$ ms | 0.3467-↑7.7% | 1.449-↑0.3% |
| controller | $\tau = 30$ ms | 0.3545-↑10.1% | 1.447-↑0.2% |

*4.2. Virtual Prototype Validation for the IVES*

A virtual prototype (VP) enabled the proof testing before assembling the hardware, reducing both the manufacturing cost and time. The design possibilities of the IVES can be explored through the VP, and the study of tradeoffs between IVES component sizes becomes feasible. In this part, an VP, combining CATIA, ADAMS, and MatLab/Simulink environment was constructed to establish a high-fidelity multi-body model for the IVES.

The build process used to create the VP model is shown in Figure 12. Firstly, a complete vehicle model was parametrically modeled in CATIA, using the vehicle model parameters obtained from an actual IWMD EV. Then, the IVES model was imported into the ADAMS and the constraints of each component were established. The collision model of each component was established, the corresponding material was defined, and the loads and drives were added. Finally, the driving torque and controller force of the ASS was taken from the IWM system modules and ASS controller modules available in MATLAB/Simulink, respectively. For the UEMF, the vertical force $F_u$ was applied to the IWM rotor and stator surfaces.

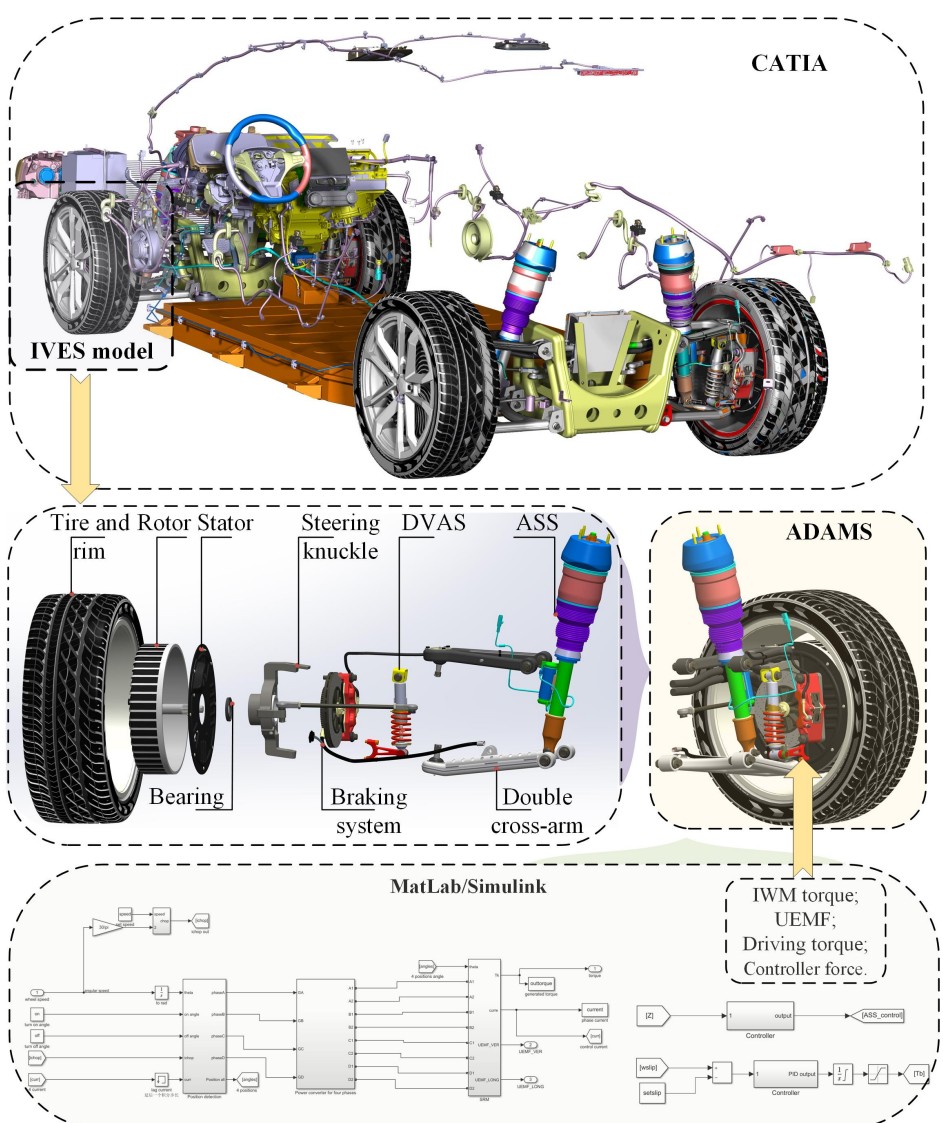

**Figure 12.** The VP building process.

As shown in Figure 12, using a connector, the DVAS is linked to the steering knuckle at the upper end and the IWM stator at the lower end. Since the axle is kept in steering synchronization with the IWM, there is only relative vertical motion between the two. The lower ASS end is fixed to the lower cross arm of the double cross arm; the control force is output through the ASS sensor. The proposed IVES structure only requires original chassis changes to the steering knuckle, while the springs and damping in the DVAS are integrated into one component. In other words, the structure requires minimal modification cost and is easy to integrate. Moreover, the built tire model is non-linear and can generate tangential forces and moments in the plane of the contact patch, as well as transient effects.

The mathematical model (MM) was validated using the VP model under ISO-B at 40 km/h. The proposed model was validated by comparing the four system responses. The responses of the MM model and the VP model of IVES are compared in Figure 13. Error statistics of the response variables is obtained to compare the two models, which is listed in Table 4. The modeling error is defined as:

$$\eta = \left| \frac{(RMS_{VP} - RMS_{MM})}{RMS_{VP}} \right| \times 100\% \tag{51}$$

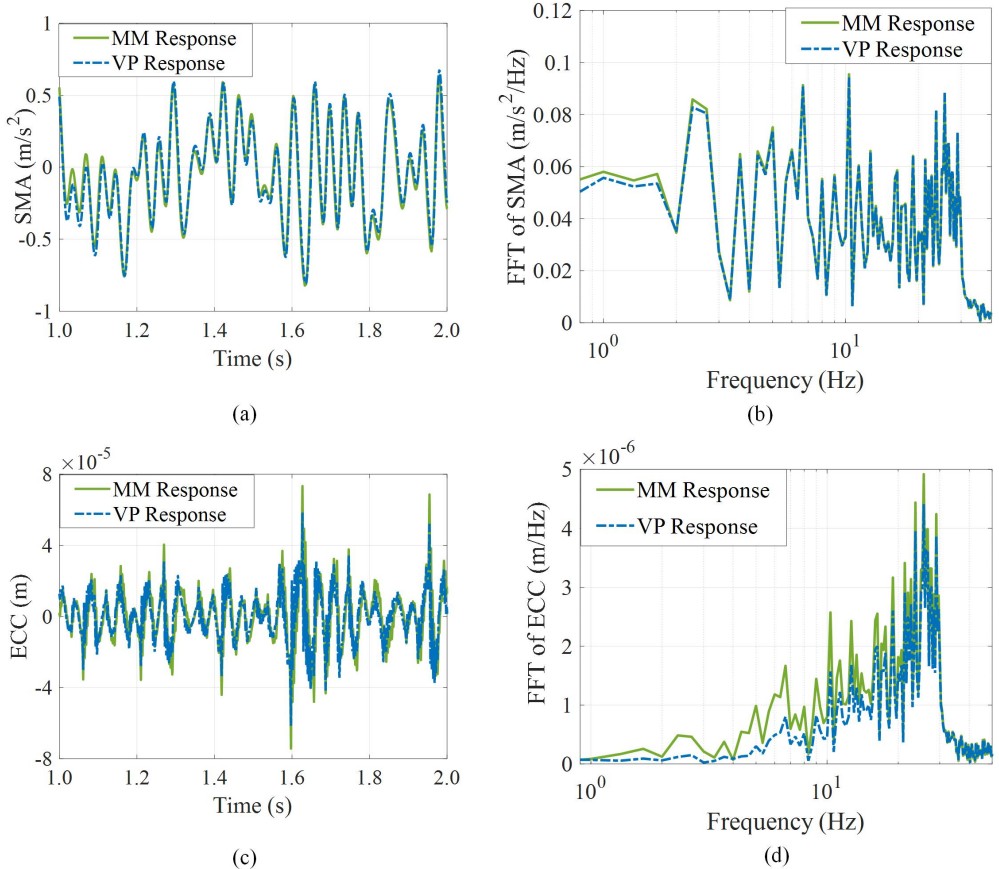

**Figure 13.** Responses of the MM and the VP: (**a**) time response of the SMA; (**b**) frequency response of the SMA; (**c**) time response of the ECC; (**d**) frequency response of the ECC.

**Table 4.** Modeling errors between the mathematical model and the VP model.

| RMS | SMA (m/s²) | ECC ($10^{-5}$ m) | RS ($10^{-3}$ m) | TD ($10^{-3}$ mm) |
|---|---|---|---|---|
| MM | 0.3217 | 1.444 | 4.222 | 4.318 |
| VP | 0.3089 | 1.323 | 4.142 | 4.039 |
| Error $\eta$ | 4.12% | 9.12% | 1.9% | 6.9% |

In the time domain, the VP model response is smaller than that of the MM for both the SMA and the ECC, with the feature being more obvious in SMA. Regarding the FFT comparison, the difference is noticeable at lower frequencies. When considering differences in the time and frequency domains, those might be caused by the tire rubber properties and the nonlinearity of the DVAS connector and the double cross arm.

Comparative results shown in Table 4 indicate that responses used in both models have an error lower than 10%. Since the error is within the permitted range [56], it was concluded that the proposed model is valid.

## 5. Conclusions

In this paper, an IVES was developed aiming to improve the vertical dynamics performance of the IWMD EVs in both sprung and unsprung conditions while also considering the UEMF effects. The mathematical model of the IVES was established, containing the DVAS-ASS integrated model and the novel FCT model. Moreover, the UEMF model during the driving maneuvers was also considered in the IVES model and was based on the theoretically analyzed UEMF effects. A delay-dependent $H\infty$ controller was developed to improve the IVES performance by improving the ASS robustness to time delay. Finally, CATIA,

ADAMS, and MATLAB/Simulink were used to create a virtual prototype environment needed to validate the accuracy and the practicability of the IVES.

Regarding the IVES performance, the numerical results obtained from the proposed mathematical model are consistent with the simulation results obtained from the virtual prototype. Compared to the passive suspension conditions, the RMS of sprung mass acceleration and the eccentricity are reduced via the IVES coordinated with the delay-dependent $H\infty$ controller by up to 67.6% and 38.9%, respectively. The ride comfort is improved and the IWM vibration is suppressed, compensating for the adverse effects of UEMF on the vehicle vertical dynamics. In future work, the real time experimental validation of the IVES applied to the IWMD EVs will be conducted.

**Author Contributions:** Project administration, L.G. and J.W.; resources, L.G. and X.Z.; supervision, H.Y. and X.Z.; validation, L.G.; visualization, Z.Z.; data curation, Z.Z. and J.W.; writing—original draft, Z.Z. and X.Z.; writing—review and editing, Z.Z. and J.W. All authors have read and agreed to the published version of the manuscript.

**Funding:** This research was funded by the National Natural Science Foundation of China (Grant No. 52202457).

**Institutional Review Board Statement:** Not applicable.

**Informed Consent Statement:** The data involved in this paper do not involve ethical issues.

**Data Availability Statement:** The datasets used and analyzed during the current study are available from the corresponding author on reasonable request.

**Conflicts of Interest:** All authors certify that they have no affiliations with or involvement in any organization or entity with any financial interest or non-financial interest in the subject matter or materials discussed in this manuscript.

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
