# Peer review of "An Integrated Vibration Elimination System with Mechanical-Electrical-Magnetic Coupling Effects for In-Wheel-Motor-Driven Electric Vehicles"

_electronics, doi:10.3390/electronics12051117_

Round 1
Reviewer 1 Report
The paper presents an integrated vibration elimination system for in-wheel-motor-driven electric vehicles using a dynamic vibration absorption structure combined with a conventional active suspension system.
The authors implemented a frequency-compatible tire model integrating rigid and flexible ring models to test the proposed vibration elimination system. The paper presents in detail the mathematical model of the implemented system.
The presented simulation results prove the effectiveness of the proposed vibration elimination system.
Reviewer 2 Report
1) The authors should explain, how the vehicle and motor have been experimentally vaidated.
2) The selection of the optimization method has to be explained.
3) The adavantages of the H-infinity controller for the ASS in this study are unclear against other control methods. How the overall real-time applicability of the developed controller has been assessed?
Reviewer 3 Report
Dear Authors,
thank You for the possibility to read and review Your work: “An integrated vibration elimination system with mechanical-electrical-magnetic coupling effects for in-wheel-motor-driven electric vehicles”.
An integrated vibration elimination system with mechanical-electrical-magnetic coupling effects for in-wheel-motor-driven electric vehicles was proposed.
First, I would like to note that the article could be improved in terms of editing and checked for linguistic correctness of specialized phrases.
My general questions and comments are as follows:
1. In the article the Authors defined in-wheel-motor-drive electric vehicles as IWMD EVs. But more often is used IWM (in-wheel-motor) which is not explained precisely – more general or more deep?
2. Why only the short time 2s is used in almost quite article? Is it possible make longer simulation test or not needed but why? Only on Fig. 10 You used the time 3s for simulation time. Why?
3. While do You considering exactly the unbalanced electric magnetic force effects because in whole article is not clearly given.
4. How we could to improve the vehicle vertical dynamics performance in the sprung and unsprung state?
5. How precise is a virtual prototype (VP) in this case?
6. What exactly system do for dynamic vibration-absorbing and where is the novel in the frequency-compatible tire vehicle vertical dynamics performance?
7. How we could improve the vehicle vertical dynamics performance in the sprung and unsprung state?
8. The distortions are cause an unbalanced electric magnetic force (UEMF) or only magnetic force which further distorts the air gap distribution, exacerbating the in-wheel-motor (IWM) vibration, creating a vicious cycle mechanical-electrical-magnetic coupling effects?
9. Finally it could be good to show how the vibration level was eliminated or reduced in proposed particular points for Case 1 up to 4. How Your proposition for vibration elimination system (IVES) improve IVES accuracy with the mechanical-electrical-magnetic coupling effects UEMF with a key role.
10. What is the correspondence or between the modelling error and the vibration elimination level or how deep is the influence?
Line by line questions:
Line 29 - Is the [8] reference properly choose and given? The article shows the calculation and measurement of unbalanced magnetic pull in cage induction motors with eccentric rotors. Are really the same problems?
Line 55 - Choi et al.?
Line 485 – it is written “validated using the AP model”, but the “AP model” was not described
Line 487 – the same but for “the VP model”
Table 4. has a "odeling errors"...
I hope that my comments will help to improve the article.
Kind regards
Reviewer
